# Preparation and Optimization of Itraconazole Transferosomes-Loaded HPMC Hydrogel for Enhancing Its Antifungal Activity: 2^3 Full Factorial Design

**DOI:** 10.3390/polym15040995

**Published:** 2023-02-16

**Authors:** Eidah M. Alyahya, Knooz Alwabsi, Amal E. Aljohani, Rawan Albalawi, Mohamed El-Sherbiny, Rehab Ahmed, Yasmin Mortagi, Mona Qushawy

**Affiliations:** 1Pharm D Program, Faculty of Pharmacy, University of Tabuk, Tabuk 71491, Saudi Arabia; 2Department of Basic Medical Sciences, College of Medicine, AlMaarefa University, Riyadh 13713, Saudi Arabia; 3Department of Anatomy and Embryology, Faculty of Medicine, Mansoura University, Mansoura 35516, Egypt; 4Department of Natural Products and Alternative Medicine, Faculty of Pharmacy, University of Tabuk, Tabuk 71491, Saudi Arabia; 5Department of Pharmaceutics, Faculty of Pharmacy, University of Khartoum, Khartoum 11111, Sudan; 6Department of Pharmaceutics, Faculty of Pharmacy, Sinai University, Alarish 45511, North Sinai, Egypt; 7Department of Pharmaceutics, Faculty of Pharmacy, University of Tabuk, Tabuk 71491, Saudi Arabia

**Keywords:** itraconazole (ITZ), transferosome (TFS), antifungal activity, *Candida albicans*, optimization, stratum corneum

## Abstract

Itraconazole (ITZ) is a triazole antifungal agent characterized by broad-spectrum activity against fungal infections. The main drawback of ITZ, when applied topically, is the low skin permeability due to the stratum corneum, the outermost layer of the skin, which represents the main barrier for drug penetration. Therefore, this study aimed to prepare itraconazole as transferosomes (ITZ-TFS) to overcome the barrier function of the skin. ITZ-TFSs were prepared by thin lipid film hydration technique using different surfactants, sodium lauryl sulfate (SLS) and sodium deoxycholate (SDC). The prepared ITZ-TFS were evaluated for entrapment efficiency (EE) %, particle size, polydispersity index (PDI), zeta potential, and in vitro drug release to obtain an optimized formula. The surface morphology of the optimized formula of ITZ-TFS was determined by transmission electron microscope (TEM). The optimized formulation was prepared in the form of gel using hydroxyl propyl methyl cellulose (HPMC) gel base. The prepared ITZ-TFS gel was evaluated for homogeneity, drug content, spreadability, pH, and in vitro antifungal activity in comparison with the free ITZ gel. The prepared ITZ-TFS formulations exhibited high EE% ranging from 89.02 ± 1.65% to 98.17 ± 1.28% with particle size ranging from 132.6 ± 2.15 nm to 384.1 ± 3.46. The PDI for all ITZ-TFSs was less than 0.5 and had a negative zeta potential. The TEM image for the optimized formulation (ITZ-TFS4) showed spherical vesicles with a smooth surface. The prepared gels had good spreadability, pH, and acceptable drug content. ITZ-TFS gel showed higher antifungal activity than free ITZ gel as determined by zone of inhibition. ITZ was successfully prepared in form of TFSs with higher antifungal activity than the free drug.

## 1. Introduction

Superficial mycosis is a fungal disease that often affects the top layers of the skin, nails, and hair but can also spread to deeper tissues. Numerous microorganisms, including bacteria and fungi, commonly inhabit the human skin (normal flora) [1]. Overgrowth of any of these members of normal flora can occur as a result of many potential underlying medical conditions. *Candida albicans* is a yeast that typically lives on the skin folds such as the armpits, digestive tract, mouth, and vagina [2]. Skin infection by *Candida albicans* (candidiasis) can cause intense itching and redness [3]. The main causes of candidiasis are warm weather, tight clothing, poor hygiene, obesity, the use of antibiotics, the use of corticosteroids, and the suppressed immune system [4].

Itraconazole (ITZ) is a triazole antifungal agent that has broad-spectrum activity [5]. It works by inhibiting ergosterol synthesis by interacting with the fungal 14 alpha-demethylase substrate-binding sites, thus inhibiting the conversion of lanosterol to ergosterol [6]. This impaired ergosterol synthesis leads to deformity of the fungal membrane that enhances permeability and disrupts the integrity of the fungal cell membrane, and membrane-bound enzyme activity is changed [7]. ITZ belongs to a BCS class II which is characterized by low water solubility. It has an extremely low aqueous solubility of approximately 1 ng/mL at neutral pH and 5 µg/mL at pH 1 [8,9].

Oral preparations of ITZ including solutions and capsules as well as intravenous preparations are well-known and widely used in medicine. Recently, due to toxicity concerns associated with systemic use, ITZ is increasingly being investigated for its topical antifungal activities and many reports proved it can be successfully delivered via this route [10,11].

Conventional topical preparations such as lotions, gels, ointments, and creams are facing several challenging factors affecting the drug delivery process [12]. These factors include low lipid solubility, the high molecular weight of the drug, and the barrier properties of the skin [13]. The stratum corneum is the outer shell of the epidermis which acts as a barrier preventing the penetration of topically applied antifungal drugs [14]. The skin permeability can be enhanced by either modifying the drug formulation or skin permeation enhancers [15]. Nanotechnology-based drug delivery systems offer effective drug delivery to overcome these problems [16]. Nanotechnology is a collection of techniques including the design, synthesis, characterization, and application of structures, materials, devices, and systems by controlling the size and shape on the nanoscale, which is capable of overcoming various challenges in drug delivery [17]. Transferosomes (TFS) are highly flexible, deformed vesicles with an aqueous core surrounded by a lipid bilayer which enables them to be a good carrier for both hydrophilic and hydrophobic drugs [18]. Several studies used nanotechnology for improving the skin permeability of ITZ. Alomrani et al. developed ITZ niosomes using different nonionic surfactants, (tween, span, and Birj) to enhance its antifungal activity [19]. Passos et al. prepared ITZ-loaded nanostructured lipid carriers to improve its skin permeability and increase its localization in skin lesions associated with fungal infections [20]. Hashem et al. prepared ITZ-loaded ufasomes to enhance the skin permeability and improve its antifungal activity against *Candida albicans* [21]. 

The aim of this work was the preparation of ITZ-TFS to enhance its antifungal activity against *Candida albicans*, using different surfactants (sodium deoxycholate (SDC) and sodium lauryl sulfate (SLS)), different surfactant amounts (200 and 400 mg), and different phospholipid amounts (200 and 400 mg). Eight formulations were prepared by thin lipid film hydration technique. The prepared formulations were examined for the entrapment efficiency % (EE%), particle size, zeta potential, polydispersity index (PDI), and in vitro permeation. The best formulation was incorporated in Hydroxypropyl Methylcellulose (HPMC) gel base and evaluated for spreadability, pH, drug content, and in vitro antifungal activity, in comparison with HPMC gel base loaded with free ITZ.

## 2. Materials and Methods

### 2.1. Materials 

Itraconazole (ITZ) was purchased from Sigma-Aldrich (St. Louis, MO, USA). Phospholipid 90 H was kindly donated by Lipoid GmbH (Ludwigshafen, Germany). Sodium deoxycholate (SDC) and sodium lauryl sulfate (SLS) were purchased from Loba Chemie (Mumbai, India). Hydroxyl propyl methyl cellulose (HPMC) was purchased from Alpha Chemica (Mumbai, India). All other chemicals were of analytical grade.

### 2.2. Design of ITZ-TFS Formulations Using 2^3 Full Factorial Design 

Design Expert software is designed to help with the design and interpretation of multi-factor experiments [22]. It offers a wide range of analytical and graphical techniques for model fitting and interpretation. A 2^3 full factorial design is a type of experimental design that designs 8 formulations and allows researchers to understand the effects of three independent variables (each of 2 levels) on the dependent variable.

In this study, design expert version 11 (Stat-Ease, Inc., Minneapolis, MN, USA) was used to obtain the composition of ITZ-TFS formulations. The independent variables were type of surfactant (X1), amount of surfactant (X2), and amount of phospholipid (X3) while the dependent variables were the entrapment efficiency % (Y1), particle size (Y2), and zeta potential (Y3). 

The type of surfactant (X1) was used in two levels (SLS and SDC), the amount of surfactant (X2) was used in two levels (200 and 400 mg), and the amount of phospholipid was used in two levels (200 and 400 mg). The independent and dependent variables of ITZ-TFS formulations are represented in Table 1. 

### 2.3. Preparations of ITZ-TFS Formulations Using Thin Lipid Film Hydration Technique 

In a dry, round-bottom flask, accurate amounts of phospholipid, surfactant, and drug were dissolved in a mixture of organic solvents (10 mL) consisting of chloroform and methanol (1:1, *v*/*v*). To prepare a thin lipid film on the wall of the round-bottom flask, the organic solvent was allowed to evaporate using a Heidolph rotavap (P/N Hei-AP Precision ML/G3, Schwabach, Germany) set to 60 rpm at 45 °C under low pressure [23]. The dry thin lipid film was hydrated with 10 mL of distilled water by rotating it at 60 rpm for 1 h at room temperature. The formed lipid vesicles were allowed to swell at room temperature for 2 h [24]. The transferosomal dispersions were then collected and sonicated, in a water bath sonicator, for 2 min then kept in a refrigerator at 4 °C overnight to complete maturation for further investigation.

### 2.4. Determination of the Entrapment Efficiency % (EE%) of ITZ-TFS Formulations 

The centrifugation method was used to determine the EE% of ITZ in the prepared ITZ-TFSs. To separate the entrapped ITZ from the unentrapped medication, 2 mL of each formulation was centrifuged at 14,000 rpm for 45 min in a cooling centrifuge (Biofuge, primo Heraeus, Germany) [25]. Using a UV spectrophotometer (Shimadzu, Kyoto, Japan), the supernatant was examined spectrophotometrically for the presence of un-entrapped ITZ at 262 nm after dilution to 100 mL with phosphate buffer saline (pH 7.4). The following equation was used to compute the EE%: $$\left(EE\%\right)=\frac{Total\;ITZ-Unentrapped\;ITZ}{Total\;ITZ}\times 100$$

### 2.5. Measurement of the Particle Size, Zeta Potential, and Polydispersity Index (PDI) of ITZ-TFS Formulations

Particle size, zeta potential, and polydispersity index were measured for the ITZ-TFS formulations that were prepared. Zeta potential is the value of charge in the boundary between the nanoparticle and the dispersion medium, the higher the value of zeta potential, the higher the physical stability of the dispersion. PDI is a value, which indicates the homogeneity or heterogeneity of particle size. Each ITZ-TFS sample was diluted to 1% concentration before being evaluated by Zetasizer (Malvern Instruments Ltd., Malvern, UK) at 25 °C. The measurement was done at a 90° angle using the dynamic light scattering technique [26].

### 2.6. In Vitro Drug Release of ITZ-TFS Formulations

The ITZ-TFS formulations’ in vitro release study was conducted utilizing Franz’s diffusion cell apparatus (Maharashtra, Mumbai, India). The donor compartment was filled with 2 mL of each ITS-TFS formulation, and the receptor compartment was filled with 12 mL of phosphate buffer saline (pH 7.4). Throughout the experiment, the dissolving medium was stirred at 100 rpm and maintained at 37 ± 1 °C [15]. The donor and receptor chambers were separated by a cellophane membrane (1.7 cm diameter). At predetermined intervals, 1 mL samples were obtained (1, 2, 4, 6, 8, 12, and 24 h). After the necessary dilutions, all samples were subjected to spectrophotometric examination using a UV spectrophotometer at 262 nm. The experiment was conducted in triplicate, and the mean ± SD was computed.

### 2.7. Selection of the Optimized Formulation of ITZ-TFS

The best ITZ-TFS was selected to complete further studies. The best formulation was selected based on the highest EE%, smallest particle size, and highest zeta potential. The optimization process was done using Design Expert software version 11 (accessed on 7 February 2023) to obtain the optimized formulae with the optimum responses. 

### 2.8. Transmission Electron Microscopy (TEM) Image of the Optimized ITZ-TFS

A transmission electron microscope (TEM), (JEOL^®^, Tokyo, Japan) was used to analyze the surface morphology of the best formulation (ITZ-TFS). After the optimal formulation was diluted with distilled water, one drop was deposited on a collodion-covered copper grid. Uranyl acetate solution was used to stain the sample, and a TEM image was captured after the stain solution had dried at room temperature [27]. TEM image was captured by TEM camera at the acceleration voltage of 100 kV and 10–100 k magnification power [28]. 

### 2.9. Preparation of ITZ-TFS Gel 

Gels were prepared using 5% HPMC as a gelling agent. The gelling agent was dispersed in distilled water overnight to prevent the formation of clumps during preparation. The gels were prepared by the addition of 0.1% (*w/w*) of the drug (either free ITZ or ITZ-TFS) with continuous stirring using a magnetic stirrer to obtain homogenous dispersion. Methyl and propyl parabens were used as preservatives [29]. 

### 2.10. Evaluation of the Prepared ITZ Gels 

Free ITZ gel and ITZ-TFS gel were examined for their homogeneity, spreadability, pH, drug content, in vitro permeation study, and in vitro antifungal activity. 

#### 2.10.1. Homogeneity 

The homogeneity of semisolid dosage forms that are placed topically on the skin is critical for patient compliance. This was accomplished by pressing a small amount of the prepared gels (free ITZ gel, and ITZ-TFS gel) between the thumb and index finger. The consistency was evaluated to determine whether it was homogeneous or not [24].

#### 2.10.2. Spreadability

The spreadability of the prepared gels (free ITZ gel and ITZ-TFS gel) was evaluated by pressing 0.5 g of each gel formulation between two circular glass slides. The lower slide was fixed while the upper one was movable [30]. Constant pressure was applied above the upper slide for 5 min to allow maximum spreading of each gel. The experiment was done in triplicate and the mean ± SD was calculated. 

#### 2.10.3. pH Determination

One gram of each prepared gel was mixed with 20 mL of distilled water and then the pH value was determined by the digital pH meter [31]. The measurements were done in triplicates and the mean ± SD was calculated. 

#### 2.10.4. Drug Content %

The drug content was determined by transferring 100 mg of each gel (free ITZ gel and ITZ-TFS gel) into a clean volumetric flask (100 mL) and filling the remaining space with distilled water [32]. The contents were agitated for 2 h before being filtered and spectrophotometrically measured at 262 nm. The measurements were taken three times to ensure accuracy, and then the mean and standard deviation were calculated.

#### 2.10.5. In Vitro Permeation Study 

The permeation study was done using the Franz diffusion cell apparatus as mentioned before in the previous section using rat skin to separate between donor and receptor compartments [33]. The protocol of this research paper was approved by the scientific research ethics committee at the Faculty of Pharmacy, Sinai University, Arish, Egypt (approval number SU-SREC-3-01-23). The rat skin was shaved and cleaned then soaked at the diffusion medium for 2 h before the experiment. One gram of each gel (free ITZ gel and ITZ-TFS gel) was placed on the donor compartment while the receptor was filled with 12 mL of phosphate buffer saline (pH 7.4) as a diffusion medium. The soaked skin was placed between the donor and receptor where the epidermis faced up and the hypodermis faced down. The permeated amount of ITZ was calculated by taking samples of the diffusion medium (1 mL) at different predetermined time intervals (1, 2, 4, 6, 8, 12, and 24 h). The samples were analyzed spectrophotometrically at 262 nm using a UV-spectrophotometer. The experiment was done in triplicate for each gel and the mean ± SD was calculated. The amount of drug permeated at different time intervals was plotted against the time to compare the permeability of ITZ from different gels. 

#### 2.10.6. In Vitro Antifungal Activity (Zone of Inhibition)

The antifungal activity of ITZ-TFS gel was determined by the cup plate technique in comparison with free ITZ gel. Sabouraud dextrose agar was added to Petri dich inoculated with *Candida albicans* and the Petri dich was rotated to allow uniform distribution of fungi within the agar. After the solidification of agar, a sterile metallic cork borer was used to make holes. Both free ITZ gel and ITZ-TFS gel were transferred to the holes and left for 2 h to allow diffusion. Then, the plates were incubated for 24 h at 25 °C. After 24 h, the zones of inhibition were measured in mm for both gels. The experiment was done in triplicate and mean ± SD was determined.

### 2.11. Statistical Analysis

All experiments were done in triplicate and the mean and standard deviation of the data were calculated (mean ± SD). ANOVA test was used to assess the significant differences among various experimental groups. Significance was considered at *p* values less than 0.05.

## 3. Results and Discussion 

Eight formulations of ITZ-TFS were prepared by thin lipid film hydration technique and evaluated for EE%, particle size, and zeta potential to obtain the optimized formulation. See Table 2. 

### 3.1. Effect of the Formulation Factors in the EE% (Y1) of ITZ-TFS Formulations

The EE% is the percentage of drug encapsulated within the nanoparticles and considered from the important formulation factors that are required to be optimized in the transferosomes formulations. As represented in Table 3, the EE% was high ranging from 89.02 ± 1.65% for ITZ-TFS8 to 98.17 ± 1.28 for ITZ-TFS1. 

As shown in Figure 1, the EE% of the prepared ITZ-TFS formulations was affected by the surfactant type. SLS resulted in higher EE% compared to SDC. These results may be due to the lower HLB (hydrophilic lipophilic balance) of SLS (HLB = 11) than SDC (HLB = 16). The lower the HLB value, the higher the lipophilicity and the affinity to the phospholipid and hence the increase in EE% [24,34]. These results were in good agreement with Abdelmonem et al., who prepared Cinnarizine transferosomes using three different surfactants (Pluronic F-127, sodium cholate, and sodium deoxycholate), with different HLB values (22, 18, and 16, respectively). They found that EE% was increased using SDC (the lowest HLB) [35]. 

Moreover, the EE% decreased by increasing the amount of surfactant which may be due to the increase in the solubility of the drug in the aqueous phase by the increasing amount of surfactant [36]. Other researchers explained the reduction in EE% by increasing the surfactant concentration by the formation of pores in the lipid bilayer membrane which resulted in leakage of the drug [37]. The previous results were in agreement with Mayangsari et al., who prepared berberine chloride tranferosomes for transdermal use using Tween 80 as a surfactant and found that the EE% was decreased by increasing the surfactant concentration from 15% to 25% [38]. Both the type and amount of surfactant exhibited significant impact on the EE% (*p* < 0.05) as indicated in Table 4.

### 3.2. Effect of the Formulation Factors on the Particle Size (Y2) of ITZ-TFS Formulations

The particle size of transferosomes is an important parameter, which gives an indication about the skin penetrating; the smaller the size, the higher the skin penetration. The particle size of the prepared ITZ-TFS formulations was in the range of 132.6 ± 2.15 for ITZ-TFS4 to 384.1 ± 3.46 for ITZ-TFS5. As shown in Figure 2, the particle size of the prepared ITZ-TFS formulations was small when SLS was used as a surfactant and increased by using SDC. These results may be due to the difference in HLB value which resulted in the difference in the hydrophilicity and hence the reduction in surface free energy by using SLS. Moreover, the long chain length of SDC leads to an increase in the critical packing parameter, thus the vesicle size increases. It has been reported that the use of a negatively charged surfactant reduces the average particle size because the negative charge causes the complex lipid bilayer of the transferosome to be susceptible to curvature due to the attractive force between the positively charged choline group on phospholipids and the negative charge on the surfactant [39]. 

In addition, the particle size decreased by increasing the amount of surfactant, which may be due to the reduction in the interfacial tension resulting in a smaller size [40]. Another explanation for the previous results may be due to the solubilization of the surfactant within the lipid bilayer as a result of the formation of hydrogen bonding between the alkyl chain of surfactant and the polar head of phospholipid which led to a reduction of the vesicle size [41]. 

The particle size also decreased by increasing the amount of phospholipid which may be attributed to the reduction in surface free energy resulting from high hydrophobicity [34]. Similar findings were obtained by Qushawy et al., who developed miconazole transferosomes and found that the vesicle size was decreased by increasing the total lipid. As represented in Table 5, it was found that all formulation factors had a significant effect on the particle size (*p* < 0.05).

As represented in Table 3, the PDI value for all ITZ-TFS formulations was less than 0.5, which indicates the homogeneity of size distribution [42]. 

### 3.3. Effect of the Formulation Factors in the Zeta Potential (Y3) of ITZ-TFS Formulations

Zeta potential is the magnitude of the charge on the surface of transferosomes vesicles in the colloidal dispersion. The value of zeta potential indicates the degree of repulsion between vesicles and the degree of physical stability. As represented in Table 3, all prepared ITZ-TFS formulations had negative zeta potential ranging from −40.9 ± 1.66 for ITZ-TFS7 to −67.4 ± 0.82 for ITZ-TFS4. The negative charge may be attributed to the anionic phospholipid and anionic surfactants. Similar results were obtained by Balata et al., who prepared vabradine HCl transferosomes and found that the negative zeta potential was attributed to SLS and phospholipid [43]. 

As shown in Figure 3, the zeta potential value was the highest in the formulations prepared by SLS followed by SDC. These results may be due to the small size of vesicles prepared by SLS. Moreover, the negative zeta potential value was increased by increasing the amount of surfactant, which may be due to the increase in surface area due to the reduction in the interfacial tension [44]. The negative zeta potential value also increased by increasing the amount of phospholipid, which may be attributed to the increase in the negatively charged phosphate group [45]. As represented in Table 6, it was found that all factors had a non-significant effect on the zeta potential (*p* > 0.05).

### 3.4. The Release Study of ITZ

As shown in Figure 4, the release of ITZ from the prepared TFS was done in two distinct phases. The initial rapid phase may be due to the release of ITZ which adsorbed on the surface of transferosomes, while the slow second phase may be due to the slow diffusion of ITZ through the phospholipid bilayer of transferosomes [46]. It was found that the release was high in the case of using SLS as a surfactant than in case of SDC. These results may be attributed to the smaller size of the prepared TFS in presence of SLS which resulted in the large surface being subjected to the dissolution medium. Moreover, the release was increased as the amount of surfactant increased which may be due to the decrease in the interfacial tension. The cumulative amount released was decreased in formulation with a high amount of phospholipid which may be due to the increase in the diffusion path length. Similar results were obtained by El-Gizawy et al., who prepared deferoxamine-loaded transferosomes to accelerate the healing of pressure ulcers in streptozotocin-induced diabetic rats [40].

### 3.5. Optimization of the Formulation Factors 

Design expert version 11 was used to optimize the formulation factors to obtain an optimized formulation with high EE%, small particle size, and high zeta potential. As shown in Figure 5, it was found that ITS-TFS4, which was prepared with SLS, 400 mg of surfactant, and 400 mg of phospholipid, was selected as the optimized formulation with a desirability index of 0.865 and it was used to complete the study. Figure 6 shows the particle size distribution curve and zeta potential of the optimized formulation (ITZ-TFS4). Kumar et al. succeeded in obtaining an optimized formulation of itraconazole niosomes with EE% of 73.03 ± 2.39%, a particle size of 272.3 nm, and a zeta potential of −47.2 ± 0.2 [47]. Samy et al. developed itraconazole as proniosomes and were able to obtain an optimized formulation with 94.9 ± 0.36 % regarding the EE%, −30.15 ± 0.41 mV regarding zeta potential, and 340.48 ± 0.581 nm regarding the particle size [48]. 

### 3.6. The Surface Morphology of the Optimized Formulation 

As shown in Figure 7, the optimized formulation appeared spherical with a smooth surface. These results were in agreement with Demartis et al. [49]. Similar results were obtained by Hady et al., who developed nystatin transferosomes for the treatment of vulvovaginal candidiasis and examined the surface morphology by TEM, and found that the prepared vesicles were spherical [50].

### 3.7. Evaluation of ITZ-TFS Gel 

All prepared gels (free ITZ gel and ITZ-TFS gel) were homogenous. As represented in Table 7, the spreadability of free ITZ gel and ITZ-TFS gel was 4.5 ± 0.46 and 4.6 ± 0.29 cm, respectively. These results indicate the ability of prepared gels to be spread easily on the skin with low shear [51]. The pH value for the prepared gels were 6.62 ± 0.52 and 6.61 ± 0.74 for free ITZ gel and ITZ-TFS respectively. These results indicate that these gels were suitable to be applied on the skin as the skin can tolerate the pH value from 4 to 7 [52]. These results are in agreement with Parveen et al., who prepared a transferosomal gel of bifonazole to enhance its antifungal activity and found that the pH of the prepared formulations was in the range of 6.3–6.6 [53]. The drug content of free ITZ gel and ITZ-TFS gel was 98.40 ± 2.47 and 95.44 ± 1.69%, respectively, which is within the acceptable range, see Table 7. 

#### In Vitro Permeation Study

As shown in Figure 8, the cumulative amount permeated of ITZ from ITZ-TFS gel was 88.33 ± 2.41% which is higher than free ITZ gel (72.77 ± 1.62%) after 24 h. These results may be attributed to the high deformability of TFS which resulted in higher permeability of the drug through the skin [54]. Furthermore, the presence of phospholipid in the prepared TFS increased their affinity to the skin and hence improved the permeation of the encapsulated drug. Similar findings were obtained by Omar et al., who prepared lidocaine transferosomes-loaded HPMC gel base and evaluated the permeation through the skin in comparison with the free lidocaine gel and found that the cumulative amount of drug permeated was higher in the case of TFS gel [52].

### 3.8. In Vitro Antifungal Activity

As shown in Figure 9, it was found that the zone of inhibition of ITZ-TFS gel (55.1 ± 0.65 mm) was higher than that for free ITZ gel (24.3 ± 0.24 mm). These results may be attributed to the high flexibility and deformability of TFS, which facilities its penetration through the cell wall of *Candida albicans* resulting in the inhibition of ergosterol biosynthesis, which results in the lysis of fungal cell membrane and loss of integrity. The results were in agreement with Singh et al., who prepared ketoconazole transferosomal gel and found that the zone of inhibition was higher than for free ketoconazole [55]. 

## 4. Conclusions 

The authors concluded that ITZ could be successfully prepared as TFSs, achieving a high EE% that ranged from 89.02 ± 1.65% to 98.17 ± 1.28%, with particle sizes that varied from 132.6 ± 2.15 nm to 384.1 ± 3.46 nm. The PDI for all ITZ-TFS was lower than 0.5 and had a high negative zeta potential value with high physical stability. The ITZ-TFS displayed spherical vesicles as determined by TEM. The prepared gels had good spreadability, an acceptable pH, and a high drug content %. According to the results of the zone of inhibition test, the antifungal activity of ITZ-TFS gel was significantly higher than that of free ITZ gel. ITZ in the form of TFS was successfully prepared, and its antifungal activity was found to be significantly higher than that of the free drug.

## Figures and Tables

**Figure 1 polymers-15-00995-f001:**
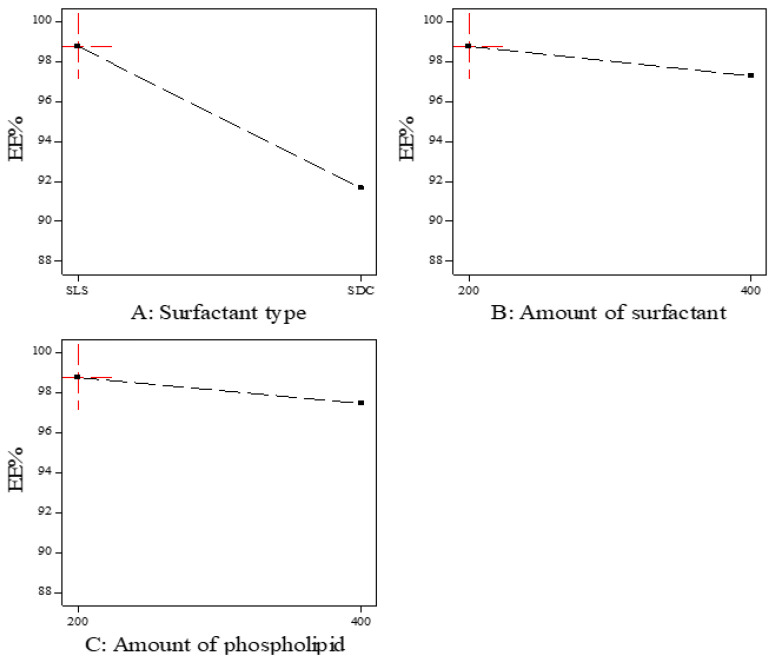
The effect of formulation factors in the EE% (Y1).

**Figure 2 polymers-15-00995-f002:**
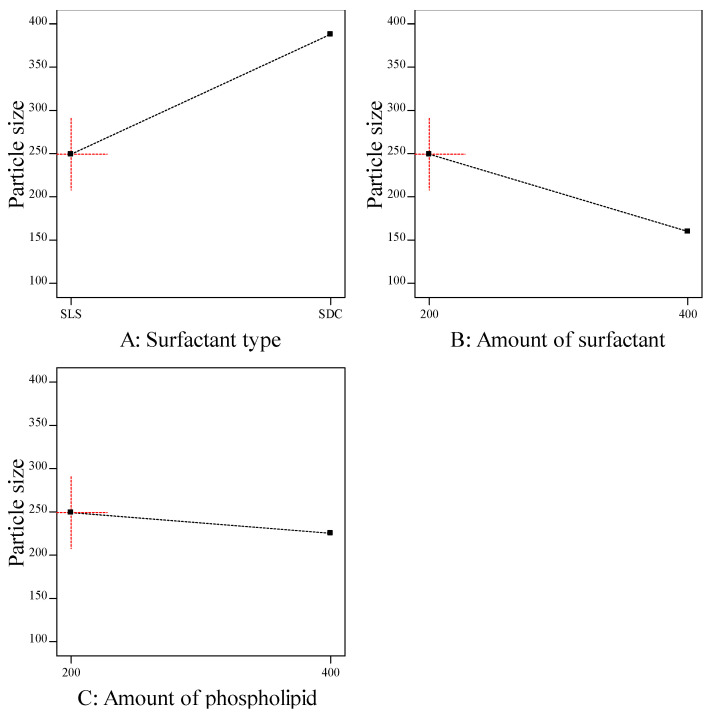
The effect of formulation factors on the particle size (Y2).

**Figure 3 polymers-15-00995-f003:**
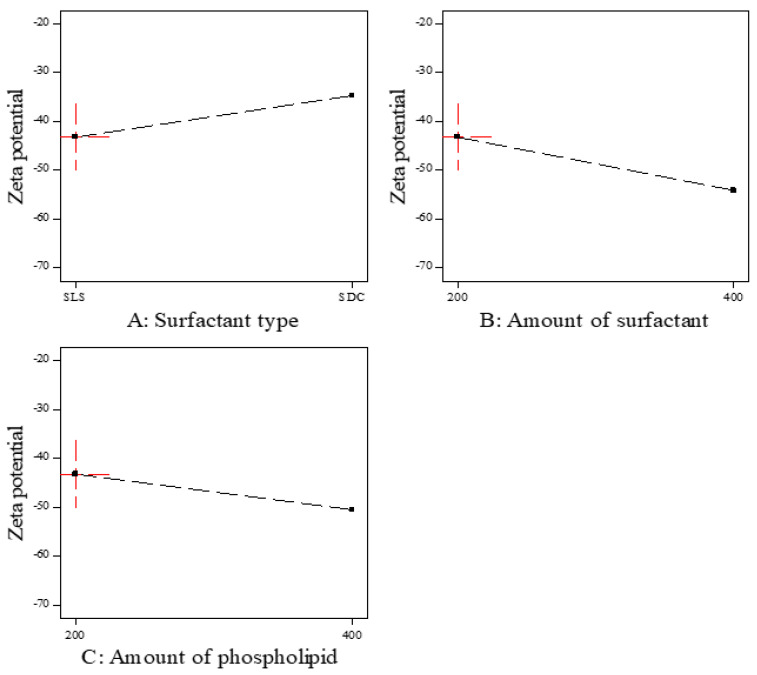
The effect of formulation factors on the Zeta potential (Y3).

**Figure 4 polymers-15-00995-f004:**
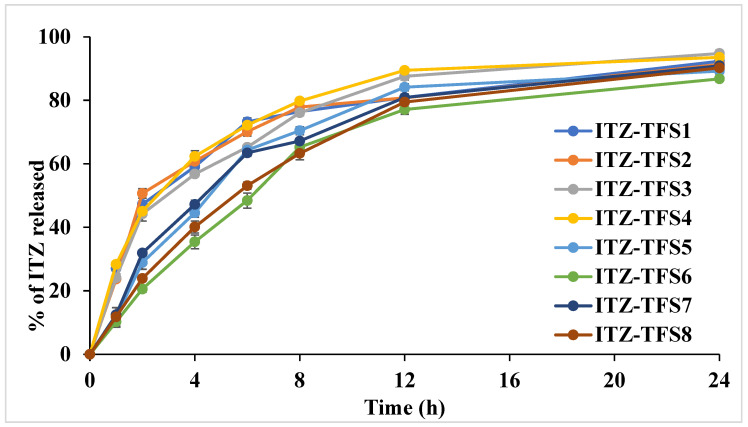
In vitro release study profile of the prepared ITZ-TFS formulations.

**Figure 5 polymers-15-00995-f005:**
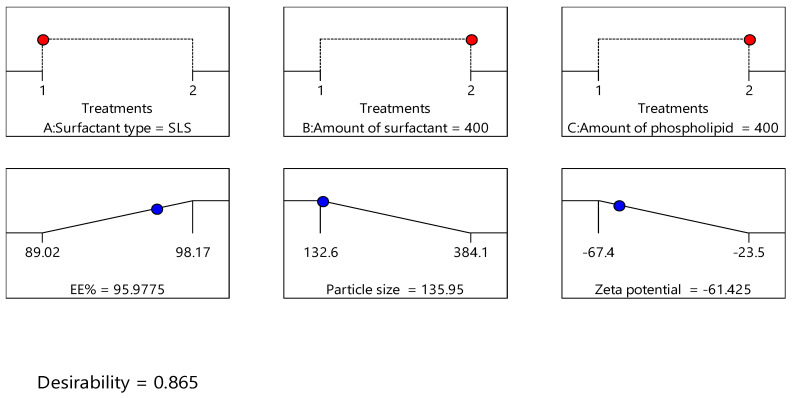
The independent (red color) and dependent variables (blue color) of optimized ITZ-TFS4.

**Figure 6 polymers-15-00995-f006:**
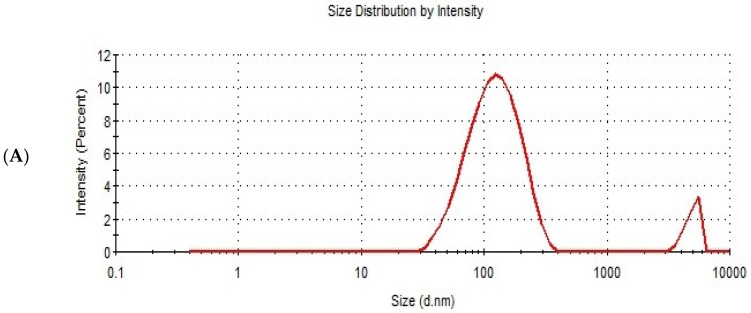
The particle size distribution curve (**A**) and zeta potential (**B**) of the optimized ITZ-TFS4.

**Figure 7 polymers-15-00995-f007:**
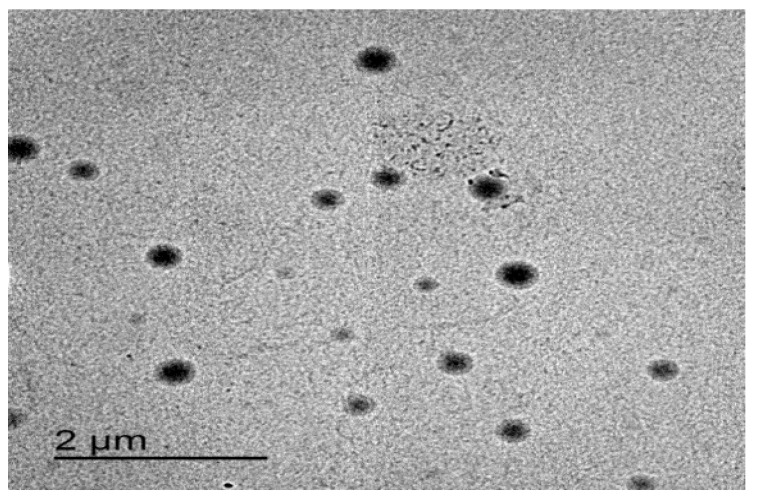
TEM image of the optimized ITZ-TFS4.

**Figure 8 polymers-15-00995-f008:**
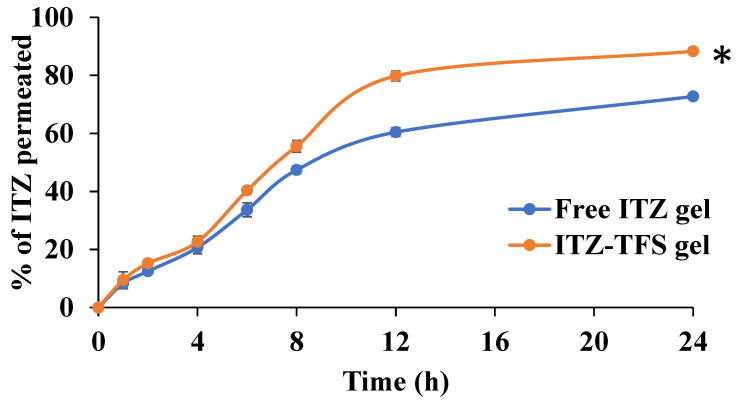
In Vitro permeation study of ITZ from the prepared gels. “*” (p ˂ 0.05) denotes a significant difference between ITZ-TFS gel and free ITZ gel.

**Figure 9 polymers-15-00995-f009:**
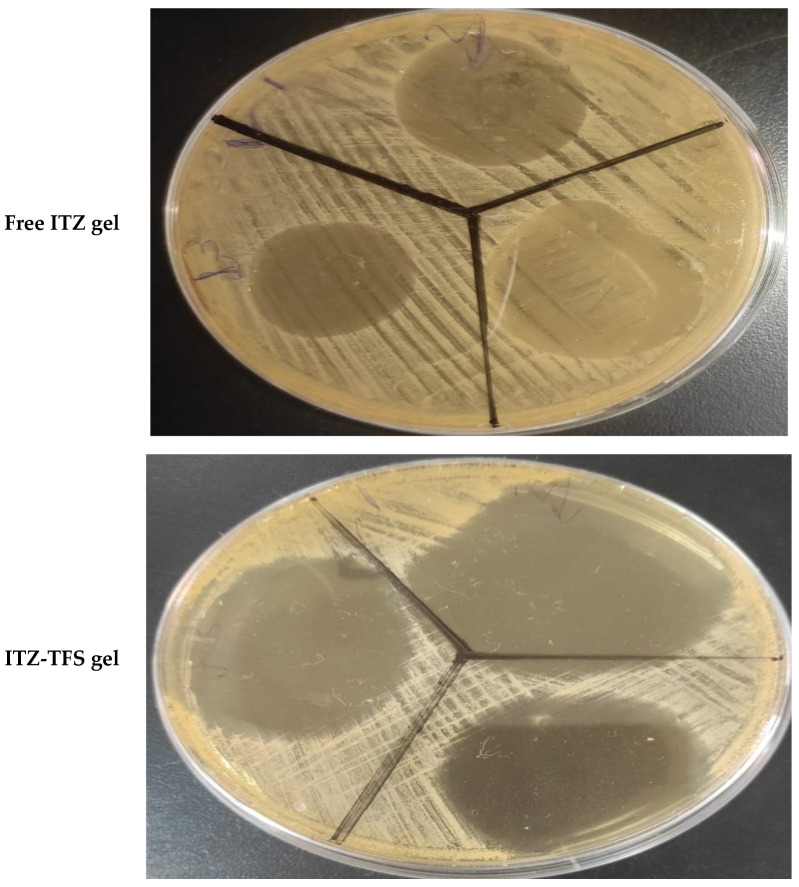
In vitro antifungal activity test showing the zone of inhibition of free ITZ gel and ITZ-TFS gel.

**Table 1 polymers-15-00995-t001:** The formulation factors and responses of 2^3 full factorial design for ITZ-TFS formulations.

Factors and Responses		Used Levels	
Factor	Name	Low (−1)	High (+1)
A: X1	Type of surfactant	SLS	SDC
B: X2	Amount of surfactant (mg)	200	400
C: X3	Amount of phospholipid (mg)	200	400
**Response**	**Name**	**Goal**	
Y1	EE (%)	**Maximize**	
Y2	Particle size (nm)	**Minimize**	
Y3	Zeta potential (mV)	**Maximize**	

**Table 2 polymers-15-00995-t002:** The designed ITZ-TFS formulations using 2^3 full factorial design.

F No.	Itraconazole (mg)	Type of Surfactant	Amount of Surfactant (mg)	Amount of Phospholipid (mg)
ITZ-TFS1	100	SLS	200	200
ITZ-TFS2	100	SLS	200	400
ITZ-TFS3	100	SLS	400	200
ITZ-TFS4	100	SLS	400	400
ITZ-TFS5	100	SDC	200	200
ITZ-TFS6	100	SDC	200	400
ITZ-TFS7	100	SDC	400	200
ITZ-TFS8	100	SDC	400	400

**Table 3 polymers-15-00995-t003:** The characterization of ITZ-TFS formulations.

F No.	EE%	Particle Size (nm)	PDI	Zeta Potential (mV)
ITZ-TFS1	98.17 ± 1.28	249.9 ± 2.01	0.468 ± 0.02	−23.5 ± 0.65
ITZ-TFS2	97.87 ± 1.75	227.9 ± 1.96	0.379 ± 0.01	−54.2 ± 0.97
ITZ-TFS3	97.27 ± 0.94	159.9 ± 3.44	0.470 ± 0.01	−64.2 ± 1.03
ITZ-TFS4	96.16 ± 0.88	132.6 ± 2.15	0.400 ± 0.02	−67.4 ± 0.82
ITZ-TFS5	92.57 ± 2.64	384.1 ± 3.46	0.426 ± 0.01	−49.01 ± 1.35
ITZ-TFS6	89.63 ± 1.76	364.4 ± 2.76	0.468 ± 0.03	−43.7 ± 0.95
ITZ-TFS7	89.87 ± 1.14	301.6 ± 4.95	0.318 ± 0.02	−40.9 ± 1.66
ITZ-TFS8	89.02 ± 1.65	274.9 ± 2.65	0.350 ± 0.01	−41.5 ± 2.01

**Table 4 polymers-15-00995-t004:** ANOVA analysis for the effect of formulation factors in the EE% (Y1).

Source	Sum of Squares	df	Mean Square	F-Value	*p*-Value	
**Model**	108.44	3	36.15	71.39	0.0006	Significant
A—Surfactant type	100.68	1	100.68	198.84	0.0001	Significant
B—Amount of surfactant	4.38	1	4.38	8.65	0.0423	Significant
C—Amount of phospholipid	3.38	1	3.38	6.68	0.0611	Not significant
**Residual**	2.03	4	0.5063			
**Cor Total**	110.46	7				

Significance was considered at *p* values less than 0.05.

**Table 5 polymers-15-00995-t005:** ANOVA analysis for the effect of formulation factors on the particle size (Y2).

Source	Sum of Squares	df	Mean Square	F-Value	*p*-Value		
**Model**	55,564.23	3	18,521.41	1745.86	<0.0001	Significant	
A—Surfactant type	38,461.51	1	38,461.51	3625.45	<0.0001	Significant	
B—Amount of surfactant	15,957.91	1	15,957.91	1504.22	<0.0001	Significant	
C—Amount of phospholipid	1144.81	1	1144.81	107.91	0.0005	Significant	
**Residual**	42.44	4	10.61				
**Cor Total**	55,606.67	7					

Significance was considered at *p* values less than 0.05.

**Table 6 polymers-15-00995-t006:** ANOVA analysis for the effect of formulation factors on the Zeta potential (Y3).

Source	Sum of Squares	df	Mean Square	F-Value	*p*-Value	
**Model**	490.40	3	163.47	0.7272	0.5869	Not significant
A—Surfactant type	146.20	1	146.20	0.6504	0.4652	Not significant
B—Amount of surfactant	237.62	1	237.62	1.06	0.3620	Not significant
C—Amount of phospholipid	106.58	1	106.58	0.4741	0.5290	Not significant
**Residual**	899.22	4	224.80			
**Cor Total**	1389.62	7				

Significance was considered at *p* values less than 0.05.

**Table 7 polymers-15-00995-t007:** Evaluation of the prepared ITZ gels.

Gel Name	Spreadability (cm)	pH	Drug Content %
**Free ITZ gel**	4.5 ± 0.46	6.62 ± 0.52	98.40 ± 2.47
**ITZ-TFS gel**	4.6 ± 0.29	6.61 ± 0.74	95.44 ± 1.69

## Data Availability

Data are available from the corresponding author upon request.

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
