# Peer review of "Preparation and Optimization of Itraconazole Transferosomes-Loaded HPMC Hydrogel for Enhancing Its Antifungal Activity: 2^3 Full Factorial Design"

_polymers, 2023, doi:10.3390/polym15040995_

Round 1

Reviewer 1 Report

Authors have prepared and optimized Itraconazole transferosomes loaded HPMC hydrogel and optimized formulations. The paper is well presented and can be accepted after checking grammatical/ spelling errors.

Reviewer 2 Report

In this manuscript, authors performed Preparation and optimization of Itraconazole transferosomes loaded HPMC hydrogel for enhancing its antifungal activity: 2^3 full factorial design  . In my opinion, some issues should be further address and I hope following comments could be helpful for improving their paper.

  1. It would be better if author add graphical representation of overall study as schematic digram.
  2. For drug release study Which PBS buffer was used?
  3. Why the size shown in TEM image is 2um?
  4. what parameters is optimized for this nanomedicine ?
  5. Is  itraconazole  is water soluble drug?
  6. It would be better to check this system on mice model.
  7. Authors need to perform biocompatibility and immuno histological analysis such as H&E staining to further confirm the effect of such nano formulation.
  8. Discussion:This part requires a thorough development. The authors should clarify the   signalized doubts. They should to demonstrate the advantages and disadvantages of the proposed system against the background of similar systems described earlier. The authors should also present their suggestions related to possible possibilities of practical application of the described solution.
  9. Please revisit the entire manuscript for minor grammar issues

Reviewer 3 Report

The presented study analyzes the incorporation of Itraconazole transferosomes in hydrogels to study its activity against Candida albicans. The work does not have much novelty in itself, in fact, the authors name other works where the same compound is loaded in different nanosystems. The following arguments should be clarified, added or discussed in order to improve the article and be taken into account for publication:

- The designed systems are based on a hydrogel which is prepared using parabens as preservatives, which are widely known to be very polluting. Therefore, the applicability of these systems in biological applications could be doubtful. In order to determine whether the system is biocompatible and does not produce any toxic effects at the biological level, the in vitro activity should also be evaluated in healthy systems.

- In section 2.2, it should be briefly explained what the design expert version 11 is.

- On line 263, it talks about "lower HLB" of SLS, but what does that mean? Nothing has been mentioned above.

- About characterization by microscopy, the power of the TEM must be indicated in section 2.8. The analysis by this technique is very scarce, only a single image is presented with very low magnification even though we are talking about nanometric systems. A size distribution histogram should be presented. The particle size data presented in Table 3, how have they been calculated, there is no image evidence of these size differences between the systems.

- The ITZ-TFS gel evaluation section is best presented as a single paragraph (or two) where a discussion of all the studied properties of these systems (spreadability, pH, drug content...) is made.

- There are many things related to the format of the article that should be taken care of: i) the terms in vitro and Candida albicans must always be in italics. ii) The term HPMC should be clarified for those who do not understand hydrogels and know their composition. iii) Many times, the references are in italics. iv) The terms Figure X and Table X are sometimes in lowercase and sometimes in uppercase. v) Some figures are of normal size, but others are too large and it is not significant that this is the case. vi) It is very important to be consistent with the number of significant figures in the tables.

Round 2

Reviewer 2 Report

Accepted in current form

Reviewer 3 Report

After the response to my comments and questions, I consider that the work may be suitable for publication.